# The Association between Quitline Characteristics and Smoking Cessation by Educational Attainment, Income, Race/Ethnicity, and Sex

**DOI:** 10.3390/ijerph18063297

**Published:** 2021-03-23

**Authors:** David C. Colston, Bethany J. Simard, Yanmei Xie, Marshall Chandler McLeod, Michael R. Elliott, James F. Thrasher, Nancy L. Fleischer

**Affiliations:** 1Center for Social Epidemiology and Population Health, Department of Epidemiology, School of Public Health, University of Michigan, Ann Arbor, MI 48109, USA; bethany.simard@gmail.com (B.J.S.); nancyfl@umich.edu (N.L.F.); 2Biostatistics Core, Rogel Cancer Center, University of Michigan, Ann Arbor, MI 48109, USA; xyanmei@umich.edu (Y.X.); marshallcmcleod@uabmc.edu (M.C.M.); 3Department of Biostatistics, School of Public Health, University of Michigan, Ann Arbor, MI 48109, USA; mrelliot@umich.edu; 4Survey Research Center, Institute for Social Research, University of Michigan, Ann Arbor, MI 48104, USA; 5Department of Health Promotion, Education, and Behavior, Arnold School of Public Health, University of South Carolina, Columbia, SC 29208, USA; thrasher@mailbox.sc.edu; 6Center for Population Health Research, Department of Tobacco Research, National Institute of Public Health, Cuernavaca 62100, Mexico

**Keywords:** smoking cessation, quitlines, disparities, tobacco control, health policy

## Abstract

Little research examines how tobacco quitlines affect disparities in smoking cessation in the United States. Our study utilized data from the Tobacco Use Supplement to the Current Population Survey (2010, 2011, 2012, 2015, 2018) (TUS-CPS) and state-level quitline data from the North American Quitline Consortium and National Quitline Data Warehouse. We ran multilevel logistic regression models assessing a state-run quitline’s budget, reach, number of counseling sessions offered per caller, and hours of operation on 90-day smoking cessation. Multiplicative interactions between all exposures and sex, race/ethnicity, income, and education were tested to understand potential effect modification. We found no evidence that budget, reach, number of counseling sessions, or hours available for counseling were associated with cessation in the main effects analyses. However, when looking at effect modification by sex, we found that higher budgets were associated with greater cessation in males relative to females. Further, higher budgets and offering more sessions had a stronger association with cessation among individuals with lower education, while available counseling hours were more strongly associated with cessation among those with higher education. No quitline characteristics examined were associated with smoking cessation. We found evidence for effect modification by sex and education. Despite proven efficacy at the individual-level, current resource allocation to quitlines may not be sufficient to improve rates of cessation.

## 1. Introduction

Smoking causes disease in almost every system in the body and is responsible for 480,000 excess deaths a year in the United States [1]. Socioeconomically disadvantaged populations suffer disproportionately from smoking due to greater use and more challenges in quitting. For example, smoking prevalence among people with less than a high school diploma is far higher than the prevalence among people with a college degree [2]. Moreover, individuals living in poverty are less likely to quit successfully, even though they attempt to quit at the same rate as the general population [3]. Notable differences also exist with respect to sex and race/ethnicity. Specifically, males smoke at higher rates than females [2], and non-Hispanic (NH) Black adults are less likely to successfully quit smoking than their NH White counterparts [4]. Further, NH Black and Hispanic/Latinx individuals who smoke are less likely than NH White individuals who smoke to utilize evidence-based treatment options such as nicotine replacement therapy (NRT) when attempting to quit [4].

State-run quitlines, which provide telephone-based counseling, self-help materials, referrals, web-based services, and free and discounted cessation medications (such as NRT) [5], have been utilized since 1992 to help smokers quit [6]. Today, quitlines are available to smokers in all 50 states, the District of Columbia, Puerto Rico, and Guam, though there is considerable heterogeneity in the extent of services offered between states [5]. Previous work has shown using quitline services helps smokers successfully quit [7,8,9], though less than 1% of all adult smokers report using these services [10]. Quitlines have the potential to help reduce tobacco-related disparities as services are free, offered over the phone, and available outside of regular business hours, making them more accessible [11,12,13]. However, evidence is mixed on how quitline utilization and effectiveness could help reduce cessation disparities. For example, one study demonstrated that individuals of lower income and educational attainment were more likely to call quitlines than their more affluent and higher educated counterparts [14], while another study found no meaningful differences in the likelihood of calling by income or education [15]. Regarding effectiveness, one study showed that individuals at the lowest level of a composite socioeconomic status (SES) were less likely to report 7-day cessation at 6 months follow-up after calling a quitline than those of the highest SES group, though this association was not significant at 3 months follow-up [7]. Other work has shown a lower prevalence of 7-day cessation at 6 months follow-up for individuals with lower levels of educational attainment relative to those with higher education [16], and a lower likelihood of cessation for Medicaid-insured populations compared to non-priority populations (individuals that are not Medicaid-insured, and also do not identify as American Indian, are not pregnant, are not younger than 18, and do not use spit tobacco) [17]. Finally, one study found no differences in cessation after calling by income or education [18].

Considerable heterogeneity exists in the extent of quitline services offered by state, which could contribute to the mixed evidence base as to how effective quitlines may be in reducing known disparities in smoking cessation. Little research has been done to analyze how specific state-level quitline characteristics impact population-level smoking cessation and tobacco-related health disparities in the United States. This study will examine (1) the association between four quitline characteristics—budget, reach, number of counseling sessions offered per quit attempt, and hours available for services—with smoking cessation and (2) whether the relationship between quitline characteristics and cessation are modified by sex, race/ethnicity, educational attainment, and income.

## 2. Materials and Methods

### Study Population

Our study population data came from three waves of the Current Population Survey Tobacco Use Supplement (TUS-CPS) (2010/2011, 2014/2015, and 2018) and were obtained through IPUMS USA [19,20,21]. The TUS-CPS is a nationally representative survey conducted biennially, as a supplementary questionnaire to the Current Population Survey (CPS) on the use of tobacco products, personal tobacco-use history, and attitudes toward tobacco use and tobacco-control policies. Additional details regarding the TUS are located elsewhere [22].

Separate analytic samples were constructed to examine the relationships between four quitline characteristics (budget, reach, counseling sessions, hours of counseling availability) and smoking cessation. Analytic samples were restricted to self-respondents who were at least 25 years old (*n* = 357,503) to capture individuals who had likely completed their education and had established smoking patterns. Subjects were excluded if they were neither current nor past-year smokers (*n* = 298,117), reported an age of initiation into smoking “fairly regularly” within two years of their current age but were missing information on smoking behavior in the past twelve months (*n* = 54), or had invalid self-response weights (*n* = 1392). Analytic samples were constructed separately by quitline characteristic due to different availability for exposure measurements by state and over time. In the budget sample, 10,978 respondents were removed due to missing budget data (final sample = 46,962 respondents). In the reach sample, 1004 were removed due to missing information regarding reach (*n* = 56,936). For the counseling sessions sample, no respondents were missing exposure data, yielding a final sample of 57,940. Finally, with respect to the counseling availability sample, 322 were missing availability data, yielding a final sample of 57,168. 

## 3. Measures

### 3.1. Outcome Variable

Our primary outcome was successful smoking cessation, which was defined as having quit for at least the past 90 days, among respondents who reported smoking 12 months prior.

### 3.2. Exposure Variables

#### Quitline Budget per Smoker

Budget per smoker was calculated by dividing the state’s quitline budget by the state adult smoking population for each year of interest. The North American Quitline Consortium (NAQC) conducts an annual state-level quitline survey. Survey budget data for fiscal years 2010/2011, 2011/2012, 2014/2015, and 2018 were linked to TUS participants by month and year of participant’s interview. State smoking population was computed by multiplying state adult population estimates from the U.S. Census Bureau [23] by the state adult smoking prevalence from the Centers for Disease Control and Prevention’s Behavioral Risk Factor Surveillance System (BRFSS) [24]. The BRFSS current adult cigarette smoking prevalence for 2011 was used to estimate the number of adult cigarette smokers for 2010, because the BRFSS methodology changed substantially in 2011, meaning that the 2010 adult cigarette smoking prevalence was not suitable for comparison with statistics of subsequent years.

### 3.3. Quitline Services Data

State-level data on reach (defined below), the number of counseling sessions offered per quit attempt, and weekly operating hours of counseling came from the National Quitline Data Warehouse (NQDW)—a repository for quarterly-reported state-level quitline services and usage data that was established by the CDC in 2010 [25]. NQDW data were matched to CPS-TUS data based on quarter and state of the respondent.

### 3.4. Counseling Sessions Offered per Quit Attempt

The number of counseling sessions offered per quit attempt to smokers meeting basic eligibility criteria for state quitlines were dichotomized into two groups: those offering fewer than 5 sessions and those offering 5 or more sessions per quit attempt. This analysis did not factor in whether quitlines offered additional sessions to callers who met certain other criteria, such as being a pregnant smoker. All missing values were the result of states not reporting data. Sixty-six missing values for counseling sessions were imputed based on neighboring non-missing quarters (quarters before and after the missing time period of interest).

### 3.5. Hours per Week Counseling Is Available

The number of hours of counseling available per week was dichotomized as states that had counseling available 7 days per week from 7 a.m. to 7 p.m., versus those that did not. Missing values were the result of states not reporting data.

### 3.6. Quitline Reach

We defined quitline reach as the percentage of smokers who received counseling and/or medication from a quitline in a given year, using the NAQC’s reach calculation as a guide [26], which has been utilized elsewhere [27]. The values for all four quarters were added together to create a year total of service recipients, which was the numerator for the reach calculation. Median state reach was determined by calculating the median of the combined reach values for each state and year. The reach exposure variable was dichotomized based on having a reach percentage above versus below median state reach.

### 3.7. Tobacco Control Policy Covariates

We used administrative data on average cigarette pack price per year from the Tax Burden on Tobacco compiled by Orzechowski and Walker and distributed by the CDC [28]. All dollar amounts were adjusted for inflation, using 2018 as the reference year. Information on smoke-free law coverage for each state by year and month was determined using data from the American Nonsmokers’ Rights Foundation [29] merged with the Census Bureau population estimates. A dichotomous variable was created to indicate whether 100% of the population was covered by smoke-free laws in both workplaces and hospitality venues (restaurants or bars) within each state.

### 3.8. Demographic Characteristics 

Individual-level sociodemographic variables come from the TUS. Sex was categorized as either male or female. Race/ethnicity was defined as individuals that identified as Non-Hispanic White (NH White), Non-Hispanic Black (NH Black), Hispanic, or multiracial or of another racial/ethnic group, which we categorized “Non-Hispanic Other.” Educational attainment was classified as having completed less than high school (<HS), high school (HS), some college, and college degree or more (college+). We classified respondents into five annual family income groups (<$15,000, $15,000–29,999, $30,000–49,999, $50,000–74,999, or >$75,000). 

## 4. Statistical Analysis

### 4.1. Weights

TUS provides survey weights that make each month of a wave representative of the US population, which is necessary in order to appropriately adjust estimates. The weighting method involves both ratio adjustment and raking using cells defined by geography (states), ethnicity (Hispanic vs. not), race (White, Black, and Other), gender, and age categories [30]. We used the self-response weight to exclude proxy respondents. We divided the self-response weight by seven for analyses, because each month is designed to be representative of the United States overall, and 7 months of data were used in the analysis.

### 4.2. Regression Models

We conducted multilevel logistic regression models with random state effects and fixed year effects, regressing 90-day smoking cessation on each quitline exposure variable in separate models, while adjusting for age, age^2^, sex, race/ethnicity, education, annual family income, state-level cigarette price, and state-level smoke-free laws. We explored differential associations between each quitline exposure variable and sex, race/ethnicity, income, and education by including interaction terms in separate models and predicting marginal probabilities of smoking cessation when the interactions were statistically significant on the multiplicative scale. Finally, we adjusted the effect modification analysis for multiple testing using the Benjamini–Hochberg correction method with the false discovery rate at 5% across the interaction models for each quitline exposure [31]. 

In sensitivity analyses, we estimated models restricted to the sample of year-ago smokers who had made a quit attempt. We also examined the relationship between each quitline exposure and successful smoking cessation for at least 30 days. Finally, we ran additional regression models not controlling for tax price or smoke-free coverage (separately), to ensure we were not over-adjusting for other tobacco control policies (results not shown). All analyses were adjusted for survey weights to account for the survey design and conducted in Stata SE, version 15 (StataCorp, College Station, TX, USA). 

## 5. Results

Table 1 provides the weighted descriptive statistics of the study samples. The average age of respondents was around 47 years, and there were more men (range of 53.8–53.9% across analytic samples) than women (46.1–46.2%) in each analytic sample. Regarding the racial/ethnic composition of the sample, 72.2–72.5% identified as non-Hispanic White, 11.9–12.2% identified as non-Hispanic Black, 10.0–10.1% identified as Hispanic, and 5.5–5.6% were categorized as non-Hispanic Other. With respect to educational attainment, 15.9–16.2% had less than a high school education, 37.8–37.9% had graduated high school, 31.6–31.9% had completed some college, and 14.3% had a college degree or more. Family income was distributed into approximate quintiles. Among year-ago smokers, approximately 8% reported cessation for 90 days or more, while 9.5% reported 30-day cessation. In the budget per smoker analytic sample, states spent on average 3.5 dollars per smoker on the quitline. In the reach sample, about 35.9% of respondents lived in a state that reached more than 0.7% of adult smokers. In the counseling sessions sample, 58.3% of the sample lived in a state whose quitline offered five or more counseling sessions per quit attempt. Finally, in the hours of counseling availability sample, 53.7% of the sample lived in a state with quitline counseling available at least from 7 a.m. to 7 p.m., 7 days per week. 

The results of the main associations between quitline characteristics and 90-day cessation are reported in Table 2. None of the four quitline exposure variables (i.e., budget per smoker, reach, counseling sessions, and counseling hours) were associated with smoking cessation in the bivariate analyses, nor in the fully adjusted regression model.

Multiplicative *p*-values for all models with interaction terms are summarized in Table 3. Before applying the Benjamini–Hochberg correction for multiple testing, we observed statistically significant interactions between quitline budget and sex, quitline budget and education, counseling sessions and education, and counseling hours and education for 90-day smoking cessation. After correcting for the 5% false discovery rate, however, the only interactions that remained statistically significant were between budget and sex and budget and education. 

We found that quitline budget was more strongly associated with a higher probability of 90-day cessation among males and more negatively associated among females (Figure 1). Additionally, there was a stronger association between quitline budget and increased likelihood of cessation among respondents with less than high school education, as compared with no association or a weak association with a lower likelihood of cessation among respondents whose education level was higher (Figure 2).

We conducted several sensitivity analyses. First, we re-estimated models for 90-day cessation after restricting to the sample of year-ago smokers who had made a quit attempt. As in the main analysis, no statistically significant main associations were observed (Appendix A
Table A1), though we did find evidence for effect modification between hours of counseling offered and cessation by educational attainment (Appendix A
Table A2). Specifically, we found offering counseling from at least 7 a.m. to 7 p.m., 7 days a week, was more strongly associated with an increased likelihood of cessation among respondents with some college, or college education or greater, compared to individuals with <HS or a HS degree (Appendix A
Figure A1). We also examined the relationship between each quitline exposure and 30-day smoking cessation (Appendix A
Table A3) and found results from adjusted models that were similar in magnitude and direction to the 90-day cessation analyses, though no interactions between sociodemographic variables and exposures were significant in predicting 30-day cessation (Appendix A
Table A4). Regression models not controlling for state-level cigarette pack price or smoke-free coverage also found no evidence for significant effects between exposures and outcomes of interest.

## 6. Discussion

Our study found no association between quitline characteristics (i.e., budget, reach, number of counseling sessions offered, and hours of availability) and 90-day cessation. However, we did find some evidence for differences in the association between budget and 90-day cessation for some sociodemographic subgroups. Specifically, the relationship between budget and 90-day cessation was significantly modified by both sex and education, with stronger associations among men than women and among individuals with less than a high school education than those with higher levels of educational attainment. 

### 6.1. Main Effects

The association between quitline characteristics and smoking cessation has not previously been studied in a nationally representative sample, so we have nothing to which we can directly compare our null findings. Over the course of our study, the mean quitline budget per smoker in our sample was $3.50 (with quitlines ranging by state and year between $0.28 and $55.60 per smoker). However, the NAQC estimated that the budget per smoker in 2018 was actually far lower, at only $1.92 [10]. This difference could be attributable to not having response data from all states or years, as well as our reduced sample subset to only past year smokers, as this could have resulted in certain state-population estimates being left-out. Both estimates, though, suggest that the majority of quitlines are vastly underfunded compared to the NAQC goal of $10.53 per smoker [10], as recommended by the CDC. Our inability to find an association with cessation may be due to the generally much lower than recommended levels of quitline funding in our sample. For budget-constrained quitlines, the NAQC has set out guidelines for maximizing impact by using the most important, cost-effective services (e.g., targeted allocations of free NRT, scaling back proactive calls), which we did not analyze [32]; future research should consider the impact of these recommendations. Nonetheless, quitlines are mostly grossly underfunded and may, therefore, be unable to effectively reach enough smokers to have a population-level impact on quitting behaviors. It is also possible that there was no discernable association due to smokers’ lack of awareness about quitlines or other obstacles to providing optimal services. Studies have shown, for instance, that media campaigns can heighten awareness and broaden a quitline’s reach [33,34] and should be prioritized when possible. 

We also found no evidence that reach was related to successful smoking cessation. No other studies have examined this specific relationship. However, a recent study examining New York’s quitline demonstrated that, while New York had an average reach of 2.9% annually from 2011 to 2016 (which was three times the national average at the time, and over four times the mean found in our study (0.7%)), the impact on smoking prevalence was relatively minimal [27]. Recent estimates from the NAQC estimated overall quitline reach to be 0.87%, 0.88%, and 0.92% in 2017, 2018, and 2019, respectively. Importantly, these values fall far short of the NAQC’s goal of reaching 6% of smokers [10], and the 8% guideline set by the CDC [12]. Previous work has demonstrated that quitline budget and reach are strongly correlated, with higher expenditures per smoker being associated with higher rates of utilization [35]. Thus, the consistently low budgets of state-run quitlines could be contributing to the relatively stagnant measures of reach over the years. Still, neither budget nor reach were shown to be significant predictors of cessation in our study, and previous studies have suggested that the budget needed to adequately fund quitlines could be prohibitive for some states [36].

Furthermore, we found no relationship between the hours available for counseling and cessation, and no literature prior to our study has directly compared the two. In an issue brief, the NAQC suggested that 24/7 operation could help reach smokers who would not otherwise participate during more conventional hours but may not be appropriate for states with insufficient demand and those not running promotional ads outside regular hours, hence not prompting greater call volume during that time [32]. Our findings may support the latter hypothesis, as we found no evidence that longer hours of operation had any benefit to improving rates of cessation.

Finally, our results showed no evidence for an association between the number of counseling sessions offered by a quitline and 90-day smoking cessation. Meta-regression analyses from a systematic review analyzing the impact of the number of calls differed from this finding, showing an association between a greater number of calls offered and a moderately higher likelihood of quitting smoking among respondents who did not call the quitline directly [8]. This discrepancy may be due to the heterogeneous nature of the study sample, as these studies included trials from countries outside of the United States. Further, 24 of 34 studies in the meta-analysis analyzing telephone-based smoking cessation counseling lines offering between 3 and 6 calls—the category most closely resembling the five call threshold defined in our study—also showed no significant differences in quit rates by the number of calls offered. Importantly, another study demonstrated that quitline callers who completed more sessions are more likely to successfully quit [37]. That being said, all participants in the study were offered five counseling sessions, yet only between 8 and 11% of callers completed all five calls [37]. This discordance suggests that the greater number of sessions offered by a quitline does not necessarily translate to greater utilization, and in turn, a higher likelihood of cessation. Some research seems to support this hypothesis, as regardless of random assignment to a 2-call protocol versus a 4-call protocol, participants completed on average one quitline call each, and only 14% of participants assigned to four calls completed more than two calls [38]. While additional research on state-specific quitline utilization rates is needed, it seems critical that quitlines focus on efforts to improve caller retention in order to realize the full potential impact on population-level smoking cessation.

### 6.2. Effect Modification

Tests for effect modification showed that the impact of quitline budget on smoking cessation was modified by sex: higher budgets were associated with an increased likelihood of cessation among males relative to females. While no literature has looked explicitly at the impact of budget on cessation, the vast majority of literature suggests that female quitline callers are less likely to successfully quit compared to males [7,18,39], though one study has shown no significant difference in cessation by sex [16]. 

We also found evidence for effect modification by educational attainment between budget and 90-day cessation in our primary analysis. Specifically, we found that the association between budget and cessation was strongest among individuals with <HS degree compared to individuals with higher educational attainment. The evidence base for differences in cessation for people of varying educational attainment after calling a quitline is mixed. One study showed that individuals with <HS or a HS degree reported 7-day abstinence at 3-months follow-up at lower percentages than those with some college or a college degree [16]. Another study found no significant difference in 7-day smoking cessation at 3 months follow-up for those of lower income or educational attainment relative to their higher SES counterparts [18]. Other studies have shown quitline calls to be associated with lower levels of cessation among individuals with Medicaid relative to non-Medicaid smokers and other non-priority populations (7-month follow-up), [17] and lower likelihoods of cessation among smokers that are of the lowest composite SES group, compared to the highest composite group at 6 months follow-up (though the relationship was not significant at 3-months follow-up) [7]. Our results with respect to differences by education in the relationship between budgets and 90-day cessation suggest that quitlines with more resources that offer a greater extent of services, could help increase the likelihood of cessation for individuals of lower SES. Given that individuals of lower SES smoke at higher rates than their higher SES counterparts, providing greater funding and services could help reduce tobacco-related health disparities.

Finally, we found no evidence for effect modification by race/ethnicity. The majority of evidence on racial/ethnic minorities quitting after calling a quitline is focused on the American Indian/Alaskan Native population [16,17,18,40]. Importantly, only one study has addressed cessation among the racial/ethnic groups analyzed in our study, and found no significant difference in 7-day abstinence at 3-months follow-up when comparing White with Black or Hispanic quitline callers [16]. Quitlines have been theorized to be particularly helpful in encouraging treatment seeking among racial/ethnic minority smokers, as they provide a virtual, more anonymous alternative for people who are uncomfortable seeking help from medical professionals due to previous experiences of racism in the form of micro-aggressions or improper care [41]. This could especially be true for Black individuals, for whom blatant racism in the medical space has been well documented [42,43,44]. Prior research has found that Black smokers who are aware of quitlines seem to call at higher rates than their White counterparts [15,41], although these findings are not uniform across years [41] and study characteristics [14]. However, Black individuals are less likely to be aware of quitlines [14] in the first place, and qualitative work shows respondents from lower SES, predominately Black counties are less likely to trust in the Quitline due to years of failed intervention strategies [45]. The same study recommends having trusted community members who were successful in quitting after calling share their story, and help get other individuals who smoke to call the quitline [45]. More research is needed to understand the potential racial/ethnic disparities in quitline awareness, utilization, and effectiveness.

### 6.3. Limitations

There are several limitations to our study. First, the repeated cross-sectional design precludes a causal interpretation, as does the lack of individual-level quitline exposure data. We attempted to account for this by controlling for the tobacco control landscape over time, adjusting for state-level taxes and smoke-free laws by year. Second, we were limited by having missing quitline data due to states not reporting, and the NQDW and NAQC omitting non-comparable data. Third, we did not analyze the impact of quitlines on individuals who smoke and have one or more mental health conditions, despite previous work which has highlighted a disparity in quit outcomes for those with mental health conditions [46,47]. More research should be devoted to understanding differences in quitline effectiveness for individuals who do and do not have mental health conditions, as well as how the extent of quitline services offered might impact these disparities. Further, more work should be done to understand the potential impact of quitlines on smoking disparities between immigrant and non-immigrant populations, as several studies have highlighted differences in smoking by immigration status, the country from which an individual immigrates, and male–female differences within groups of individuals who have immigrated, though findings as to the direction of these disparities have varied across studies [48,49]. Additionally, our study did not factor in the extent to which quitlines were advertised, nor how media campaigns might have influenced reach or cessation. Finally, operationalizing number of sessions as being those offered to “all eligible callers” and not considering subgroups may have distorted differences in sessions offered to callers based on their age, sex, or SES, and therefore may have led to erroneous associations between session offerings and cessation.

## 7. Conclusions

Our study found that quitline characteristics (i.e., budget, reach, number of counseling sessions offered, and hours available for counseling) were not associated with 90-day smoking cessation. However, we did find evidence for effect modification between budget and cessation by sex and education. The education results, in particular, suggest that investing in quitlines has the potential to improve smoking cessation among lower SES groups, thus reducing tobacco-related health disparities. More research is necessary to understand how budget-constrained quitlines are allocating their resources, and if additional funding would yield more promising results.

## Figures and Tables

**Figure 1 ijerph-18-03297-f001:**
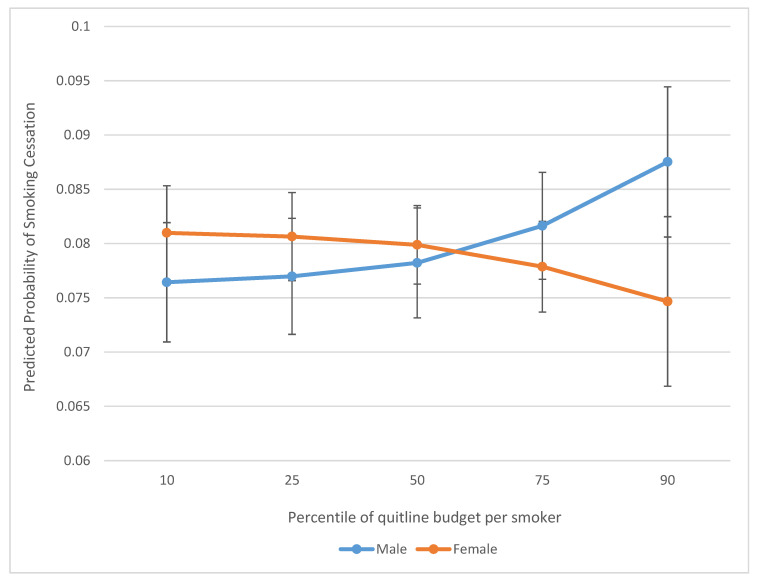
Predicted probability of 90-day smoking cessation associated with quitline budget per smoker, by gender.

**Figure 2 ijerph-18-03297-f002:**
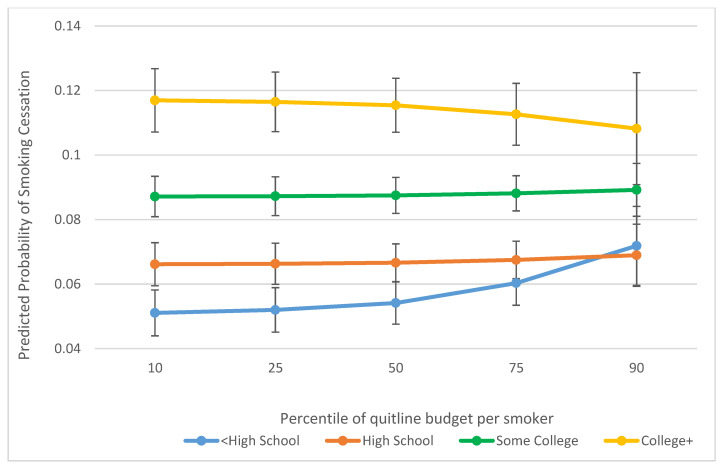
Predicted probability of 90-day smoking cessation associated with quitline budget per smoker, by education.

**Table 1 ijerph-18-03297-t001:** Weighted characteristics of quitline budget, quitline reach, counseling sessions, and counseling hours analytic samples, Tobacco Use Supplement to Current Population Survey, 2010–2018.

Variables	Quitline Budget Analytic Sample	Quitline Reach Analytic Sample	Counseling Sessions Analytic Sample	Counseling Hours Analytic Sample
Age, mean (SD)	46.5 (0.2)	46.7 (0.2)	46.7 (0.2)	46.7 (0.2)
Sex, %				
Male	53.9%	53.8%	53.8%	53.8%
Female	46.1%	46.2%	46.2%	46.2%
Race/Ethnicity, %				
Non-Hispanic White	72.2%	72.4%	72.4%	72.5%
Non-Hispanic Black	12.2%	11.9%	11.9%	11.9%
Hispanic	10.0%	10.1%	10.0%	10.0%
Other Non-Hispanic	5.5%	5.6%	5.6%	5.6%
Income, %				
$0–14,999	20.7%	20.2%	20.1%	20.1%
$15,000–29,999	21.3%	20.9%	20.8%	20.8%
$30,000–$49,999	22.8%	22.8%	22.8%	22.8%
$50,000–74,999	17.0%	17.3%	17.4%	17.4%
$75,000 and over	18.2%	18.8%	18.9%	18.8%
Education, %				
Less than High School	16.2%	16.0%	15.9%	16.0%
HS Graduate	37.9%	37.8%	37.9%	37.9%
Some College	31.6%	31.9%	31.9%	31.9%
College	14.3%	14.3%	14.3%	14.3%
90-day Cessation, %	7.975%	7.985%	7.980%	7.971%
30-day Cessation, %	9.523%	9.544%	9.547%	9.541%
State-level Smoke-free laws, %	66.9%	68.5%	68.9%	68.8%
State-level Cigarette Price, mean (SD)	6.5 (0.3)	6.6 (0.3)	6.6 (0.3)	6.6 (0.3)
N	46,962	56,936	57,940	57,618
Budget per smoker in USD, mean (SD)	3.5 (0.5)	--	--	--
Reaching ≥ 0.7% of smokers, %	--	35.9%		
Counseling sessions ≥ 5, %	--	--	58.3%	--
Counseling available at least from 7 a.m. to 7 p.m. (7 days/week), %	--	--	--	53.7%

Note. N = unweighted sample size. % = weighted percentage. SD = weighted standard deviation.

**Table 2 ijerph-18-03297-t002:** Odds ratios for 90-day smoking cessation associated with quitline exposure variables.

	Bivariate Model ^a^	Adjusted Model ^b^
	OR (95% CI)	OR (95% CI)
Budget per smoker	1.005 (0.998, 1.011)	1.004 (0.998, 1.010)
Reach	1.005 (0.932, 1.082)	0.975 (0.911, 1.043)
≥Median reach vs. <median		
Counseling sessions	0.988 (0.909, 1.075)	0.971 (0.901, 1.045)
≥5 sessions offered vs. <5		
Hours Available for Counseling 7 a.m–7 p.m. vs. not 7 a.m.–7 p.m.	1.017 (0.915, 1.130)	1.019 (0.915, 1.135)

Note. OR = odds ratio. CI = confidence interval. ^a^. Bivariate associations, with fixed year and random state effects. ^b^. Model controls for age, age^2^, sex, race/ethnicity, education, family income, state-level cigarette price, state-level smoke-free laws, with fixed year and random state effects.

**Table 3 ijerph-18-03297-t003:** Unadjusted multiplicative *p*-values ^a^ associated with interaction terms between quitline exposure variables and sociodemographic variables for 90-day smoking cessation.

	Budget per Smoker	Reach	Counseling Sessions	Hours of Counseling
Sex	**0.008**	0.429	0.585	0.452
Race/ethnicity	0.949	0.568	0.095	0.565
Education	**0.023**	0.055	0.022 *	0.022 *
Family income	0.897	0.321	0.770	0.054

Note. Each interaction is estimated from a separate model with all main effects and a single interaction term between quitline exposure and either sex, race/ethnicity, education, or income variable. All models controlled for age, age^2^, sex, race/ethnicity, education, family income, state-level cigarette price, state-level smoke-free laws, with fixed year and random state effects. ^a^. *p* values were obtained from testing multiplicative interactions and are unadjusted for multiple comparisons. Boldface numbers indicate statistically significant interactions (*p* < 0.05) after adjusting for multiple testing. Numbers with * were statistically significant (*p* < 0.05) prior to multiple testing, but not after.

## Data Availability

TUS-CPS data are publicly available at https://usa.ipums.org/usa.

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
