# Peer review of "The Association between Quitline Characteristics and Smoking Cessation by Educational Attainment, Income, Race/Ethnicity, and Sex"

_ijerph, 2021, doi:10.3390/ijerph18063297_

Round 1
Reviewer 1 Report
The manuscript is well-written and easy to understand. Methodology looks reasonable and observation is interesting. Wroth publishing.
One minor point;
p.2 Line 84: TUS should be CPS-TSU.
Reviewer 2 Report
This manuscript is about a study that looks at the correlation/association of smoking quitline parameters and various participant variables/characteristics. The subject matter is important. The manuscript is also written and articulated well. However, there is some information that I found was missing that could otherwise strengthen the paper further. The manuscript requires major changes as indicated below:
- Lines 18-19: Is CPS the acronym for Current Population Survey? If so, please place acronym in parenthesis here.
- Lines 24 to 26: You state “no evidence” that budget was “associated with cessation.” However, you later state “higher budgets were associated with greater cessation in males”. These two statements are contradictory. Please resolve. Perhaps remove “budget” from line 24?
- Line 43: The literature review is a bit scant. There is evidence that among some immigrants and racialized populations, more females reported smoking compared to males. It may be worthwhile to cite the work of researchers who show this link, See https://www.mdpi.com/1660-4601/17/17/6027/pdf . Also noteworthy may be: Azagba, S.; Sharaf, M.F. The effect of job stress on smoking and alcohol consumption. Health Econ. Rev. 2011, 15, 1–14.
- Line 54: “report calling [10].” sounds awkward. “report using these services [10].” may sound better.
- Line 76: This study “will examine” or this study examines? Has the study not been carried out yet?!
- Line 84: you don’t need “hereafter” in the parenthesis as it is obvious TUS is the acronym.
- Line 83: Is the acronym CPS? If so, please place in parenthesis.
- Line 118: you may need to write out Centers for Disease Control the first time and then use the acronym (CDC) because this is an international journal and not everyone would be familiar with North American acronyms.
- Line 167: please place the following: “(college +)” after “more” as it is not clear that it refers to a college degree or more (in the explanation for Table 1).
- Line 200: “average age” of what? Respondents?
- Table 2: I don’t understand the purpose of this table if there are no significant differences. The odds ratios are barely meaningful for me. Please respond.
If you decide to incorporate these revisions, please upload a manuscript that contains tracked changes or other method to highlight revisions. Thank you for the opportunity to review this work.
Reviewer 3 Report
A useful paper with implications for tobacco control and public health methods in general. This work begs several questions, including whether there are more effective / cost-effective methods than “quit lines” for reaching smokers and providing quit awareness and support, especially in the internet age.
I wonder whether “quit lines” were more successful back in the 1980s when smoking was more commonplace and less skewed by SES. At that time, pre-internet, “quit lines” were one of the few ways smokers could obtain information and counseling about their smoking cessation. The way we access information has changed, and perhaps “quit lines” have passed their use-by date as a mass-reach public health tool. Perhaps you could mention this in your discussion.
Another factor might be the changing demographic of smokers. In the 1970s and ‘80s smoking was relatively classless and spread across all walks of life. That is no longer the case. People who could easily give up smoking have generally done so already, and young people are simply not talking up the habit in the same numbers as 40 years ago due to successful public health messaging. So who are the current smokers? In the modern age, tobacco addiction is closely and independently associated with common mental illness like depression, anxiety and stress. People with these common mental disorders represent a high proportion of current smokers, and they may be less likely to contact a quit line.
Did you measure this among your cohorts? I’d like to see mention of the relationship between mental health and smoking in your discussion. People with mental illness, even the common types mentioned above, smoke at higher rates, smoke more cigarettes per day, smoke for longer, and have greater difficulty quitting, compared with all other smokers. There have been many papers on this topic over the past 20 years.
Round 2
Reviewer 2 Report
This manuscript is about a study that looks at the correlation/association of smoking quitline parameters and various participant variables/characteristics. The subject matter is important. The manuscript is also written and articulated well. Although the authors followed most of the reviewer’s suggestions for revision but not quite all, the responses to the remaining items are adequate. Accordingly, it is recommended that the manuscript should be published, with the following minor revision: please note that the references are listed twice (1-46; and then 1 to 50). Please keep only one list of references.
Congratulations, and thank you for the opportunity to review this work.